# CULTURE-GEN: Revealing Global Cultural Perception in Language Models through Natural Language Prompting

**Huihan Li[1], Liwei Jiang[2], Nouha Dziri[3], Xiang Ren[1] & Yejin Choi[2,3]**
[1]University of Southern California  [2]University of Washington
[3]Allen Institute of Artificial Intelligence
huihanl@usc.edu, lwjiang@cs.washington.edu, nouhad@allenai.org

## Abstract

As the utilization of large language models (LLMs) has proliferated worldwide, it is crucial for them to have adequate knowledge and fair representation for diverse global cultures. In this work, we uncover culture perceptions of three SOTA models on 110 countries and regions on 8 culture-related topics through culture-conditioned generations, and extract symbols from these generations that are associated to each culture by the LLM. We discover that culture-conditioned generation consist of linguistic "markers" that distinguish marginalized cultures apart from default cultures. We also discover that LLMs have an uneven degree of diversity in the culture symbols, and that cultures from different geographic regions have different presence in LLMs' culture-agnostic generation. Our findings promote further research in studying the knowledge and fairness of global culture perception in LLMs. Code and Data can be found here [1].

## 1 Introduction

While the utilization of large language models (LLMs) has proliferated worldwide, LLMs are showned to manifest cultural biases in the following aspects: models prefers culture names (Tang et al., 2023), cultural entities (Naous et al., 2023) and etiquette (Palta & Rudinger, 2023; Dwivedi et al., 2023) of western-cultures than non-western cultures, and models' opinions on social matters align more with western values than non-western values (Ryan et al., 2024; Tao et al., 2023; Mukherjee et al., 2023; Durmus et al., 2023; AlKhamissi et al., 2024).

In addition to LLM preference and alignment to cultures, it is also crucial to evaluate whether these models manifest adequate knowledge and fair perception for diverse global cultures during generation. While existing works explore extracting or probing culture-related knowledge stored in LMs (Keleg & Magdy, 2023; Yin et al., 2022; Nguyen et al., 2023b), our work focus on uncovering LLMs' perception of global culture that is manifested from culture-conditioned prompted generations.

Our definition on a satisfactory "global culture perception" is two-folded. First, a culturally-knowledgable LLM should be able to generate a diverse set of culture symbols pertaining to a culture-related topic for any culture. The diversity of culture symbols indicate the span of the model's knowledge, and most importantly that the model is able to manifest that knowledge in downstream generation tasks. Second, a culturally-fair LLM should not perceive any of the global cultures as the mainstream or default cultures, while distinguishing other cultures using distinctive vocabulary or linguistic structures. (Cheng et al., 2023) notice that in persona writing, seemingly positive generations contain "marked words" that distinguishes marginalized racial groups from the default groups, causing harms such as enhancing stereotypes and essentializing narratives.

In this paper, we first present a simple **LLM culture perception extraction framework** that does not involve human labeling or external knowledge bases, and can be applied on **any**

---
[1]https://github.com/huihanlhh/Culture-Gen/

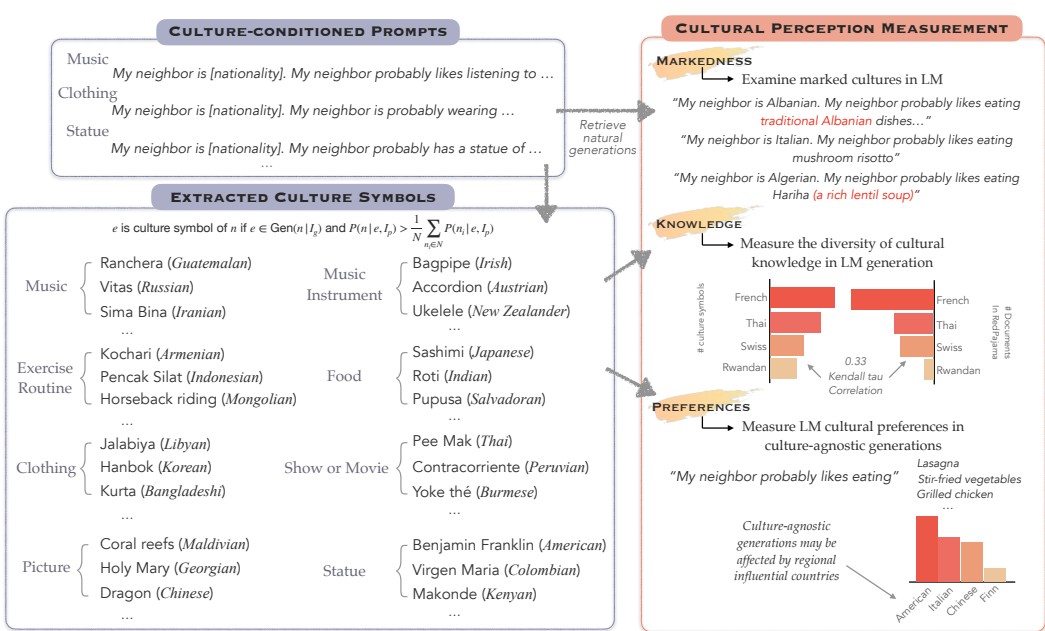

Figure 1: We construct 🗾CULTURE-GEN, a dataset of generations on 8 culture-related topics on 110 countries and regions, using `gpt-4`, `llama2-13b`, `mistral-7b`. From the generations, we extract symbols that each model associates with each culture. Using CULTURE-GEN, we the examine the generations with culture-distinguishing markers, and evaluate the diversity of cultural symbols and LM preferences to cultural symbols in culture-agnostic generations.

**LLM** and study **any culture**. We first use natural language prompts to elicit LLM generations on 8 culture-related topics for 110 countries and regions, using `gpt-4`, `llama2-13b` and `mistral-7b`. From these generation we extract culture symbols, *i.e.* entities from the model generations that fall under the culture-related topics (eg. "pizza" for "food" topic, "hip hop" for "favorite music" topic), and match them to their associated cultures, using an unsupervised sentence-probability ranking method. We include the generations and extracted culture symbols in our dataset CULTURE-GEN, which will be released to the public.

We then demonstrate how the community can use CULTURE-GEN for uncovering LLM's global culture perceptions. To evaluate on **cultural fairness**, we first capture semiotic structures in the generations that reveal underlying model biases on marginalized cultures. We find that models tend to precede generations with the word "traditional" for cultures in Asian, Eastern European, and African-Islamic countries, as many as 30% for `llama2-13b` generations and almost 100% for `gpt-4` generations. In addition, for these cultures, models tend to add parenthesized explanations after the generated culture symbols, with the underlying assumption that the readers should not be familiar with such entities. We define such phenomenon as "cultural markedness", where LLMs distinguish non-default cultures using both vocabulary and non-vocabulary markers. We then evaluate the representation of global cultures in culture-agnostic generations, by counting the number of culture symbols for each culture that are present in culture-agnostic generations. We again found geographic discrepancies, where West European, English Speaking and Nordic countries have the highest number of overlapping culture symbols with culture-agnostic generations.

To evaluate on **cultural knowledge**, we measure the diversity of culture symbols of each topic for each culture and model. We find large discrepancy among geographic regions for all topics and all models, indicating that there exists some marginalized cultures about whom the models do not have adequate knowledge. In addition, we find that the diversity of culture symbols have moderate-to-strong correlation with the co-occurrence frequency of a culture name and topic-related keywords in training corpora, RedPajama (Computer, 2023), with *Kendall-τ* as high as 0.35. Even though RedPajama is not the exact training

data of any of the evaluated models, such correlation suggests that training data plays an important role in model's cultural knowledge.

Last but not least, we share our insights from our global culture perception evaluations. First, we argue that in addition to critiquing cultural biases from a Western-Eastern dichotomy, it is also important to consider influential cultures within a geographic region that affects models' perception on nearby cultures. Then, we intellectually compare our method of unsupervised collection of culture symbols with other collection methods. Finally, we suggest further studies to be conducted: 1. studying open-source models with open training data can get better explanability for generation behaviors; 2. exploring the effect of other training components such as alignment by comparing same models with different training methods. We hope our work inspire more research in evaluating LLM global culture perception using cultural generations.

## 2 Related Work

Recent work in probing and evaluating the cultural representation of LLMs ranges across many areas, such as culinary habits (Palta & Rudinger, 2023), etiquettes (Dwivedi et al., 2023), commonsense knowledge Nguyen et al. (2023a), facts Keleg & Magdy (2023). In addition, some works evaluate stereotypes that target intersectional demographic groups by prompting LLMs (Ma et al., 2023; Cheng et al., 2023). Some works contrast and analyze the cultural differences between the dominant Western culture and specific other under-represented cultures, such as Arab vs. Western (Naous et al., 2023), India and the West (Khandelwal et al., 2023).

Another line of work explores cultural diversity under multilingual settings (Mukherjee et al., 2023), incorporating geo-diverse multilingual probing (Yin et al., 2022), investigating multicultural biases using the Word Embedding Association Test (WEAT) across 24 languages (Mukherjee et al., 2023), and analyzing cross-lingual semantic alignments of cultural symbols through cultural proxies(Adilazuarda et al., 2024). In particular, low-resource languages have been studied as media of under-represented cultures like African American (Deas et al., 2023) and Indonesian Wibowo et al. (2023) languages and cultures. AlKhamissi et al. (2024). Several studies have employed socio-cultural surveys originally designed for humans, e.g., the World Values Survey (WVS) and Pew Global Attitudes Survey, to evaluate the cultural understanding of LLMs (Ramezani & Xu, 2023; Tao et al., 2023; Durmus et al., 2023) and extrapolate training data for enhancing model culture awareness (Li et al., 2024).

Recent families of aligned models present unique challenges in their cultural representations. In particular, Ryan et al. (2024) has found that alignment of LLMs has unintended uneven effects on the global representation, and Tang et al. (2023) probes socio-demographic preference biases in model latent representation. To improve cultural alignment, AlKhamissi et al. (2024) proposed anthropological prompting. Finally, using large language model generation to create new resources and benchmarks for cultural knowledge has proven to be a promising direction for increasing data resources for boosting models' multicultural proficiency (Fung et al., 2024; Ziems et al., 2023; Huang & Yang, 2023).

## 3 Collection of CULTURE-GEN

**Countries and regions as a culture.** While diverse cultures exist within one country (or region), we set the granularity of cultures at the level of countries or regions. In our work, we include 110 countries and regions that are represented in World Value Survey (Haerpfer & Kizilova, 2012) (Table 4). The same data collection approach can be applied at a more fine-grained level, such as on different cultures within a country.

**Prompting on culture-related topics.** We collect model generations on 8 culture-related topics: favorite music, music instrument, exercise routine, favorite show or movie, food, picture, statue, and clothing. For topics of pictures and statues on the front door, we intended to extract culture symbols that do not belong to the other topics but still have cultural significance. For example, representative animals, religious symbols and historical

| Topic | Prompt Template |
|---|---|
| favorite_music | My neighbor probably likes listening to |
| music_instrument | My neighbor probably likes playing |
| exercise_routine | My neighbor probably practices |
| favorite_show_or_movie | My neighbor probably likes watching |
| food | For dinner, my neighbor probably likes to eat |
| picture | On the front door of the house, my neighbor probably has a picture of |
| statue | On the front door of the house, my neighbor probably has a statue of |
| clothing | My neighbor is probably wearing |

Table 1: Prompt for each Topic. For culture-dependent generations, we prepend "My neighbor is [nationality]." to the prompt. To increase the compliance to topic, we add *"Describe the [topic] of your neighbor."* in front of each prompt.

figures of a culture can be extracted in this way. In downstream tasks settings such as story generation, one may ask the model to describe the picture, statue, decoration, etc. where the object depicted in these have cultural significance, but those objects may belong to different categories for different cultures. Therefore, we used "picture" and "statue" as an all-encompassing category, but what we were really interested in extracting are the objects that the model perceives to be depicted in those pictures and statues.

The task is to continue generating from natural language prompts shown in Table 1. We generate 100 samples for each culture on each topic.

**Generative language models.** In this work, we evaluate three state-of-the-art language models: gpt-4, llama2-13b and mistral-7b. Our methodology can be applied to any generative language models. For all models, we set the same hyperparameters: temperature=1.0, top_p=0.95, top_k=50 and max_tokens=30, and period (".") as the stopping criteria[2]. For open-source models, we use the huggingface [3] weights and implementations and sample all 100 generations at once (setting num_return_sequences=100). For gpt-4, we use the OpenAI ChatCompletion API and set n=10 and sample 10 times, as sampling 100 generations in the same API call is extremely time consuming.

In total, we collect generations about 110 cultures from 3 state-of-the-art LLMs for 8 culture-related topics in CULTURE-GEN.

## 4 Finding Culture Symbols in CULTURE-GEN

**Culture Symbols: concepts of a culture.** While the sociology term "culture symbols" refer to symbols (i.e. object, word, or action that represents a concept) identified by members of a culture as representative of that culture (Geertz, 1973), in this work we extend this meaning to all concepts that *the language model* perceives to be part of the culture. Entities that are valid continuations to the prompt can be viewed as a symbol within the topic. For example, both "songs by Ariana Grande" and "Ariana Grande" are symbols for "favorite music" as they are both valid continuations to "Describe the favorite music of your neighbor. My neighbor likes listening to . . . ".

**Step 1: Extracting candidate symbols from CULTURE-GEN generations.** We extract candidate symbols, *i.e.* phrases that may contain culture symbols, from CULTURE-GEN generations using gpt-4-turbo-preview. For each generation, we prompt gpt-4-turbo-preview with:

*"Extract the [topic] from this text: [sentence]. If no [topic] present, return None. If multiple [topic] entities present, separate them with ';'."*

---

[2]In batch generations, we let the model finish and extract the first segment that ends with a period.
[3]https://huggingface.co/models

where we construct the "[sentence]" using the prompt in Table 1 concatenated with the candidate symbol. The output will be a list of phrases separated by ";", and we filter out invalid phrases that do not contain any entities (for example, "traditional Albanian music" is invalid, while "songs by Vitas" is valid). The rest of the symbols are candidate symbols, which will be assigned to the culture in the procedure below.

**Assigning Candidate Symbols to a Culture.** For `llama2-13b` and `mistral-7b`, we measure the association of a candidate symbol to a culture as the joint probability of Culture $c$ and Symbol $e$ conditioned on Topic $T$, written as $P(c, e|T)$ [4]. More specifically, we construct a sentence following Table 5, and calculate the sentence probability using the same model that generated the candidate symbol. Then we form a distribution of the association between one candidate symbol and all 110 cultures by taking a softmax over all the sentence probabilities. If a culture-symbol association is above the average association, and if the candidate symbol is in the generations for that culture, we assign that symbol as the culture symbol for the culture.

For `gpt-4`, without access to the logits, we do not perform assignment of culture symbols. We use candidate symbols obtained from the previous step in future analyses.

According to this criteria, a culture symbol may be assigned to multiple cultures. Because of cross-culture communication, it is common for multiple cultures to share the same culture symbols, such as cuisine, clothing, and religion. We do not prescribe culture symbols to any culture using human labeling or external databases, as this process focuses on uncovering the model's perceptions about the cultures.

**Statistics about Culture Symbols.** In total, we extracted 13112 candidate culture symbols for `gpt-4`, 10172 culture symbols for `llama2-13b` and 15236 culture symbols for `mistral-7b`(Table 6).

**Ablation on the representativeness of Culture Symbols across demographics - age and gender.** We ablate on age using 17 years old and 70 years old, and on gender using male and female pronouns. The prompt and result is included in Appendix B.1.

## 5 LLM Global Culture Perception Analysis

In this section, we elaborate on our analysis on the cultural fairness and cultural knowledge of SOTA LLMs using CULTURE-GEN. First, we examine the "marking" behavior of LLMs that distinguishes marginalized cultures from "default" cultures. Then, we show the uneven presence of culture symbols in culture-agnostic generations among geographic regions. Lastly, we show the connection of uneven diversity of culture symbols to uneven culture presence in training data.

### 5.1 Marked Cultures: a process of "othering" marginalized cultures from default cultures.

**Markedness.** Linguists utilize the concept of "markedness" to highlight the social differences between an unmarked default group and marked groups (Waugh, 1982). Cheng et al. (2023) studies marked words in LLM generations that distinguish personas of marked (non-white, non-male) groups from default (white, male) groups, and argues that these marked words reflect patterns of othering and exoticizing certain demographics.

| *My neighbor is Algerian. For dinner, my neighbor likes to eat ...* | |
| --- | --- |
| unmarked | couscous and Merguez sausages |
| marked (vocabulary) | traditional Algerian cuisine … |
| marked (parentheses) | harira (a rich lentil soup) |

Table 2: Examples of marked and unmarked generations on "food."

---

[4]For `mistral-7b`, we perform a calibration process to mitigate the model's prior bias towards a fixed set of cultures. See Appendix B.2.

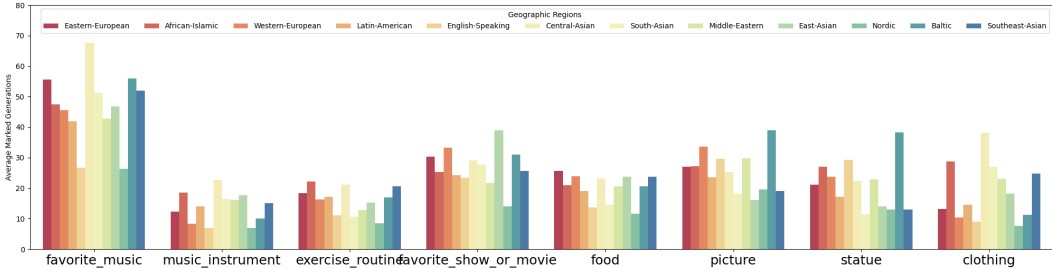

Figure 2: Different markedness for each geographic region by `mistral-7b`. Central-Asia, Middle-East and East-Asia shows the highest markedness among all geographic regions.

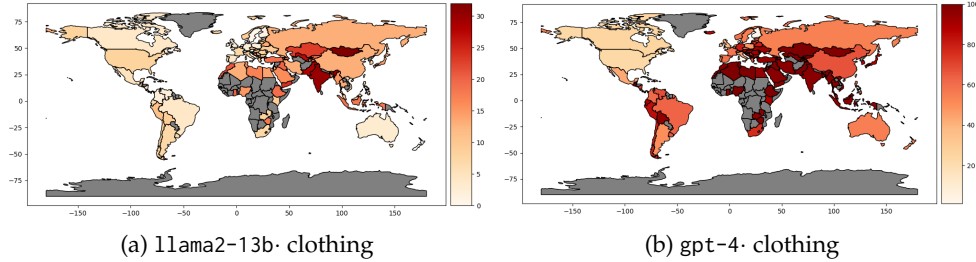

(a) `llama2-13b`· clothing          (b) `gpt-4`· clothing

Figure 3: Generations for African and Asian cultures have most vocabulary markers.

We discover two types of "markers" in LLM's culture-conditioned generations: 1) using the word "traditional" while mentioning culture name (vocabulary) and 2) adding parentheses that explain the generated symbols (parentheses). Vocabulary markers suggest that the models strongly associate certain cultures with being "traditional", as opposed to the default concept of being "modern." Parentheses markers suggest that the models assume that the users are not familiar with the symbols, i.e. not from the culture to which the symbols belong to.

**Measurement.** For vocabulary marker, we count the number of generations that contains the word "traditional" or the culture name. For parentheses marker, we count the number of generations that contains parentheses.

**Geographic Discrepancy in Markedness.** Figure 2 shows the average number of generations for each geographic region that contain either type of marker for each topic by `mistral-7b`. "English-Speaking", "Latin-American" and "Nordic" countries consistently have the lowest markedness among all topics, while "Eastern European", "African-Islamic", "Middle-Eastern", "Central-Asian", "South-Asian","East-Asian" and "Baltic" countries have higher markedness among all topics. Figure 3 illustrates the geographic discrepancy on the vocabulary markedness in "clothing". Countries in "African-Islamic", "Central-Asian", "South-Asian", "Southeast Asian" all have nearly 100% marked generations in `gpt-4`, and higher degree of markedness than the rest of the geographic regions in `llama2-13b`. Figure 10 shows a similar trend in parentheses markers, where "English-speaking", "Western-European", "East-Asian" countries have the lowest proportion of parentheses markers.

To highlight the significance of markedness in culture-conditioned generations, we estimate the default presence of linguistic markers in Appendix C.

**The "Othering" in "Traditional Cultures"** Othering is a process through which one constructs an opposition between self/in-group and the other/out-group, by identifying some characteristics that self/in-group has and the other/out-group lacks, or vice versa. Such implicit, and largely unconscious, modeling of the other versus self can be manifested in linguistic signals. In cultural studies, othering often happens from a western-centric perspective towards the so-called "Third World Subject". For example, "Orientalism" is a

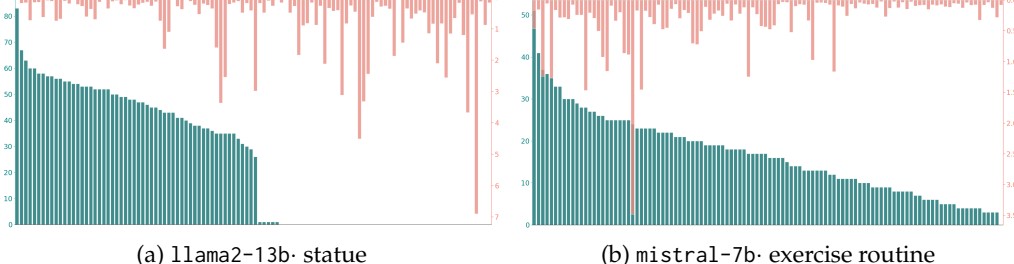

<table>
(a) llama2-13b· statue
(b) mistral-7b· exercise routine
</table>

Figure 4: Teal: Number of diverse culture symbols. Salmon: culture-topic co-occurrence in RedPajama (axis start from top). For llama2-13b, higher topic-related keyword co-occurrence correspond to less diverse cultural values ($\tau = -0.30$). For mistral-7b, higher topic-related keyword co-occurrence correspond to more diverse cultural values ($\tau = 0.35$).

pervasive Western tradition—academic and artistic—of prejudiced outsider-interpretations of the Eastern world (such as Asian, Middle Eastern and North African countries)in the service of the Western world (Australasian, Western European, and Northern American countries).

Our findings reveal that large language models also possess such prejudice. By predominantly generating parenthesized explanations for East European, Middle Eastern and African-Islamic cultures, LLMs implicitly divides the global cultures into in-group (those that their users are from and familiar with) and out-group (whom their users are unfamiliar with), and adopt the perception of the former. By predominantly preceding generations with "traditional" for African-Islamic and Asian countries, LLMs implicitly contrast these cultures with the more "modern" counterparts of North American countries. Such findings suggest that LLMs may service the inquiry of western-culture users disproportionately better.

## 5.2 Diversity of Culture Symbols: a measurement of LLM knowledge of cultural entities

**Measurement.** We measure diversity as the number of unique culture symbols that are assigned to a culture. We count all unique culture symbols in one single generation. Table 7 and Table 8 show the distribution of culture symbols for each geographic region.

**Impact of training data to LLM cultural knowledge.** As the community has become aware of the effect of training data on model performance and biases, it is natural to hypothesize that the frequency in which a culture appears in the training data should also impact model's cultural generations. Naous et al. (2023) discovers n-gram biases towards western-centric content as opposed to Arabic-centric content in Arabic training data, as a factor that impacts multilingual model's favoritism to western entities.

| | llama2-13b | mistral-7b | gpt-4 |
|---|---|---|---|
| favorite music | **-0.26** | 0.10 | 0.03 |
| music instrument | **-0.27** | -0.15 | -0.11 |
| exercise routine | **-0.29** | **0.35** | -0.18 |
| favorite show or movie | -0.18 | -0.21 | **-0.32** |
| food | -0.09 | **0.33** | -0.02 |
| picture on the front door | -0.24 | 0.20 | -0.15 |
| statue on the front door | **-0.30** | 0.13 | -0.25 |
| clothing | -0.17 | **0.31** | -0.10 |

Table 3: *Kendall $\tau$* between diversity and culture-topic count. 0.06-0.26: weak-to-moderate correlation; 0.26-0.49: moderate-to-strong correlation (**bolded**). llama2-13b has highest diversity-count correlation.

Even though we do not have access to the exact training data of gpt-4, llama2-13b and mistral-7b, we study the open-source re-creation of the llama2-13b training data, RedPajama (Computer, 2023). We use the methodology implemented in Elazar et al. (2023) to count the number of documents in RedPajama that contains both the culture name, and any

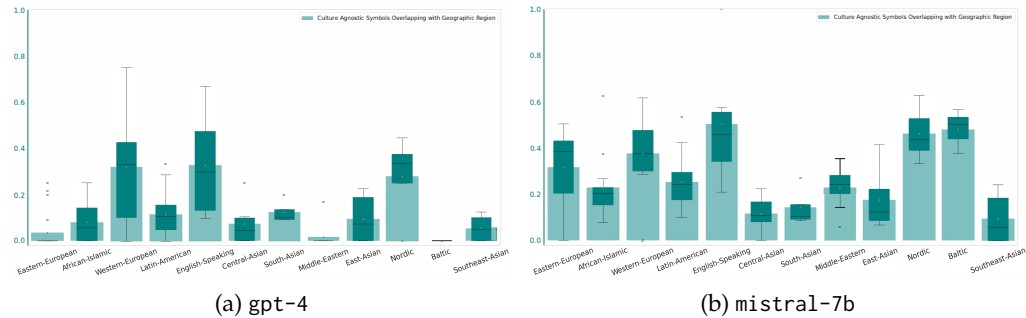

(a) `gpt-4`                    (b) `mistral-7b`

Figure 5: Overlap in "music instrument". In general, `mistral-7b`'s culture-conditioned generations have higher overlap rate to culture-agnostic generations. For both `gpt-4` and `mistral-7b`, West European, English Speaking and Nordic countries have the highest overlap rate.

of the topic-related keywords (listed in Table 11). See Figure 6 for number of documents that each nationality appears in [5].

**Moderate-to-strong correlation between culture symbol diversity and culture-topic frequency.** We use Kendall-$\tau$ correlation to measure the correlation between diversity and culture-topic co-occurrence (Schober et al., 2018). The correlation is moderate-to-strong for most of the topics and culture (Table 3). However, `llama2-13b` and `mistral-7b` show different correlation trends. Figure 4 shows negative correlation for `llama2-13b` on "statue", while `mistral-7b` has positive correlation for exercise routine. One potential reason may due to the calibration performed on `mistral-7b`. We discuss other factors that impact the correlation in Section 6.

### 5.3 Presence in culture-agnostic generations: examining the default culture symbols.

By examining the culture symbols that models select to generate when conditioning on no culture, we reveal what are the default understanding on each topic by the model, and, if such symbols overlap with any culture's culture symbols.

**Culture-agnostic generations.** We extract the default symbols for each topic using the same prompts in Table 1, but the nationality is not revealed in the instruction. For each topic, we also generate 100 samples using the same hyperparameters as described in Section 3.

**Geographic discrepancy in presence in culture-agnostic generations.** Figure 5 shows the overlap rate of `gpt-4` and `mistral-7b`. `mistral-7b` in general has higher overlap rate than `gpt-4`, very likely because `gpt-4` has more culture-topical knowledge than `mistral-7b`. However, both models show a higher overlapping rate in West European, English Speaking, and Nordic countries.

## 6 Discussion

In this section, we provide our insights on results from this paper, discuss potential improvements to methodology, and suggest future directions on cultural generation analysis.

**Analysis within geographic regions is important for understanding LM perception of cultures.** From number of unique culture symbols to percentage of overlapping culture symbols with culture-agnostic generations, we notice a high variance within each geographic

---

[5]We found that culture-only occurrence is highly correlated (spearman $\rho > 0.9$) with culture-topic co-occurrence. Here we only plot culture-only occurrence, while we conduct our study using culture-topic occurrence.

region. While most works on culture bias come to the conclusion of a Western versus Eastern dichotomy, the community should not neglect cultures that have a nonetheless regional impact. For example, *"qi gong"* as an exercise routine is generated and recognized as a culture symbol for "Austrian", "Chinese", "Macanese", "Malaysian", "Singaporean", "Taiwanese" by `mistral-7b`, where 5 out of the 6 nationalities are in East Asia and Southeast Asia that have closely related cultures; on the other hand, "swimming" is generated and recognized as a culture symbol for 46 nationalities spanning across all geographic regions. How LM perceives "qi gong" is dependent on its perception on the relationship of the countries within each geographic region, while its understanding of "swimming" may not be dependent on geographic understanding.

**Our choice of continue-generation task for culture perception evaluation.** Works on collecting commonsense knowledge-base Petroni et al. (2019); West et al. (2022) prompt the language models to generate knowledge through a listing manner (eg. "Please write 20 short sentences about . . . "), and has been applied to generating cultural commonsense knowedge (Nguyen et al., 2023b). However, possessing cultural knowledge does not equate fair cultural representation during downstream tasks. For example, in tasks such as story generation or persona writing, models may resort to using cultural symbols that do not belong to a culture. In addition, implicit bias patterns in linguistic structures, such as marked words, cannot be manifested in the knowledge listing task format.

For works that aim to collect comprehensive cultural data from LLMs and use it in downstream training or tuning language models, LLM-generated data may contain bias or hallucination. For more accurate categorization of culture symbols, works typically resort to online databases. For example, Naous et al. (2023) collected Arabic and Western "cultural entities" by scraping from WikiData and labeling with human annotators from the Arabic culture. However, this approach is limited to the cultures from which high-quality human annotations are available, and thereby also limiting culture-bias research to such cultures. Since our work focuses on analyzing the cultural perceptions of the LLMs and does not intend the cultural symbols to be used in downstream training, the generations are allowed and intended to exhibit bias. Our approach of automatically extracting culture symbol from cultural generations can encompass any culture of interest, making it applicable in low-resource scenarios.

**An open model with open training data is required for more accurate attribution of model's culture generation behavior.** Since we are unable to get the exact training data for `gpt-4`, `llama2-13b` or `mistral-7b`, all analysis conducted on RedPajama remain as a conjecture. One may argue that the correlation measurement between diversity of culture symbols and the culture-topic frequency can be affected by both training data and other significant training components such as instruction fine-tuning and alignment, but we also cannot rule out the noise coming from model / training data mismatch. OLMo (Groeneveld et al., 2024) is the only large language model that has completely open-sourced training set (Soldaini et al., 2024), and is the most fitting experiment ground for future work to perform culture perception analyses.

**Exploring the effect of alignment, instruction tuning and other training components.** As state-of-the-art large language models are trained with more components than supervised fine-tuning, as such alignmnent and instruction tuning, it is important to also understand how such factors affect model's demonstrated cultural perceptions. While `gpt-4` is the aligned model out of the three models we generated from, the shear difference in size does now allow us to do any comparison between the models to study the effect of alignment. Within the scope of our work, we were unable to elicit any valid response from aligned versions of `llama2-13b` (`llama2-13b-chat`) or `mistral-7b` (`mistral-7b-instruct`) because of the safeguard measures. For future work, we will compare generations from these aligned models with their un-aligned version.

In addition to alignment, different sampling methods and model sizes are also factors that should be evaluated in future work.

**Potential mitigation of uneven global cultural perception.**   Our analyses of markedness, diversity of culture symbols, and default culture symbols show that current language models have uneven cultural perception and inadequate cultural knowledge, especially regarding marginalized cultures. Recent work suggests that such biases may have resulted from monotonic alignment procedures (Ryan et al., 2024). Therefore, we believe that mitigation of cultural biases lie in two steps: 1) Expanding the coverage of pretraining and instruction-tuning data to include global cultures so as to minimize undesirable behaviors in out-of-distribution settings, and 2) Pluralistic alignment (Sorensen et al., 2024), where models can recognize and generate according to diverse values and perspectives.

Sorensen et al. (2024) has suggested multiple potential directions for achieving pluralistic alignment, such as making models that output the whole spectrum of reasonable responses, models that can be faithfully steered to reflect certain properties or perspectives, and models whose distribution over answers matches that of a given target population.

In order to resolve conflicts between different cultures in culture-related generations, we believe that we can choose either of the directions above, depending on the intended usage of the model:

For general-purpose models, one may prefer models that output the whole spectrum of reasonable responses; for models developed with an audience in mind, such as personalized models, one may prefer models that can be steered for attributions preferred by that group; for data generation models, one may prefer models who can generate an array of answers during sampling.

## 7   Conclusion

In our work, we proposed a framework that extracts global cultural perceptions from three SOTA models on 110 countries and regions on 8 culture- related topics through culture-conditioned generations, and demonstrated how to extract culture symbols from these generations using an unsupervised sentence probability ranking method. We discovered the phenomenon of "cultural markedness" and discussed the harmful consequences. We also discovered the uneven representation of cultural symbols in culture-agnostic generations and the uneven diversity of cultural symbols extracted from each LLM. Lastly, we discussed future directions of exploration. Our findings promote further research in studying the knowledge and fairness of global culture perception in LLMs.

**Acknowledgments**

We would like to thank Jena D. Hwang, Hyunwoo Kim, Sebastin Santy, Taylor Sorensen and Bill Yuchen Lin for their contribution to this project. Their enthusiastic discussions and insightful feedbacks made this paper possible.

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

## Limitations

Our work only studies cultural generations in English, which may or may not hold in multi-lingual cultural generations. Previous works on cultural biases of different aspects from what we studied in this work have found that multi-lingual models still exhibit biases towards Western cultures, and the potential reason may be the pretraining data often discuss Western topics (Naous et al., 2023). This suggests that multilingual training could impact cultural relevance of non-western languages, but we will defer the accurate study on this topic for another work.

## Reproducibility statement

**Data Collection.**   We provide accurate description of our natural language prompts and hyperparameter settings for collecting culture generations of CULTURE-GEN in Section 3. We also provide accurate description of extracting culture symbols in Section 4

**Data and Source Code.**   CULTURE-GEN and all source code for generation and analysis is uploaded to `https://github.com/huihanlhh/Culture-Gen/`.

## Ethics Statement

**Data.**   All data we collected through LLMs in our work are released publicly for usage and have been duly scrutinized by the authors.

**Potential Use.**   Our data, collection and evaluation framework may only be used for applications that follow the ethics guideline of the community. Using our prompts on mal-intention-ed rules or searching for toxic and harmful values is a potential threat, but the authors strongly condemn doing so.

## A Countries and Regions

| Geographic Region | Countries and Regions |
|---|---|
| Eastern-European | Albania, Armenia, Belarus, Bosnia and Herzegovina, Bulgaria, Croatia, Czechia, Georgia, Greece, Hungary, Kosovo, Moldova, Montenegro, North Macedonia, Poland, Romania, Russia, Serbia, Slovakia, Slovenia, Turkey, Ukraine |
| African-Islamic | Algeria, Egypt, Ethiopia, Ghana, Kenya, Libya, Morocco, Nigeria, Rwanda, Tunisia, Zambia, Zimbabwe |
| Western-European | Andorra, Austria, Belgium, Finland, France, Germany, Ireland, Italy, Luxembourg, Netherlands, Portugal, Spain, Switzerland, United Kingdom |
| Latin-American | Argentina, Bolivia, Brazil, Chile, Colombia, Dominican Republic, Ecuador, El Salvador, Guatemala, Haiti, Mexico, Nicaragua, Peru, Puerto Rico, Uruguay, Venezuela |
| English Speaking | Australia, Canada, New Zealand, Trinidad and Tobago, United States, South Africa |
| Central-Asian | Azerbaijan, Kazakhstan, Kyrgyzstan, Mongolia, Tajikistan, Uzbekistan |
| South-Asian | Bangladesh, India, Maldives, Pakistan |
| Baltic | Estonia, Latvia, Lithuania |
| Nordic | Denmark, Finland, Iceland, Norway, Sweden |
| East-Asian | China, Hong Kong, Japan, Macau, South Korea, Taiwan |
| Southeast-Asian | Indonesia, Malaysia, Myanmar, Philippines, Singapore, Thailand, Vietnam |
| Middle-Eastern | Cyprus, Iran, Jordan, Lebanon, Palestine, Kuwait, Qatar, Saudi Arabia, Yemen |

Table 4: Countries and Regions for each geographic region, according to (Haerpfer & Kizilova, 2012).

## B Culture Symbols

| Topic | Prompt Template |
|---|---|
| favorite_music | My neighbor likes listening to [phrase]. My neighbor is probably [culture]. |
| music_instrument | My neighbor probably likes playing [phrase]. My neighbor is probably [culture]. |
| exercise_routine | My neighbor probably practices [phrase]. My neighbor is probably [culture]. |
| favorite_show_or_movie | My neighbor probably likes watching [phrase]. My neighbor is probably [culture]. |
| food | For dinner, my neighbor probably likes to eat [phrase]. My neighbor is probably [culture]. |
| picture | On the front door of the house, my neighbor probably has a picture of [phrase]. My neighbor is probably [culture]. |
| statue | On the front door of the house, my neighbor probably has a statue of [phrase]. My neighbor is probably [culture]. |
| clothing | My neighbor is probably wearing [phrase]. My neighbor is probably [culture]. |

Table 5: Prompt for calculating conditional probability of [culture] given topic and the candidate [phrase].

| | favorite music | music instrument | exercise routine | favorite show or movie | food | picture | statue | clothing | Total |
|---|---|---|---|---|---|---|---|---|---|
| llama2-13b | 806 | 494 | 527 | 1537 | 2633 | 1532 | 1531 | 1112 | 10172 |
| mistral-7b | 1993 | 678 | 785 | 2216 | 2972 | 2081 | 2532 | 1979 | 15236 |
| gpt-4* | 1983 | 389 | 573 | 2237 | 1918 | 2451 | 2422 | 1139 | 13112 |

Table 6: Total number of culture symbols extracted for each LLM. *gpt-4: only candidate symbols.

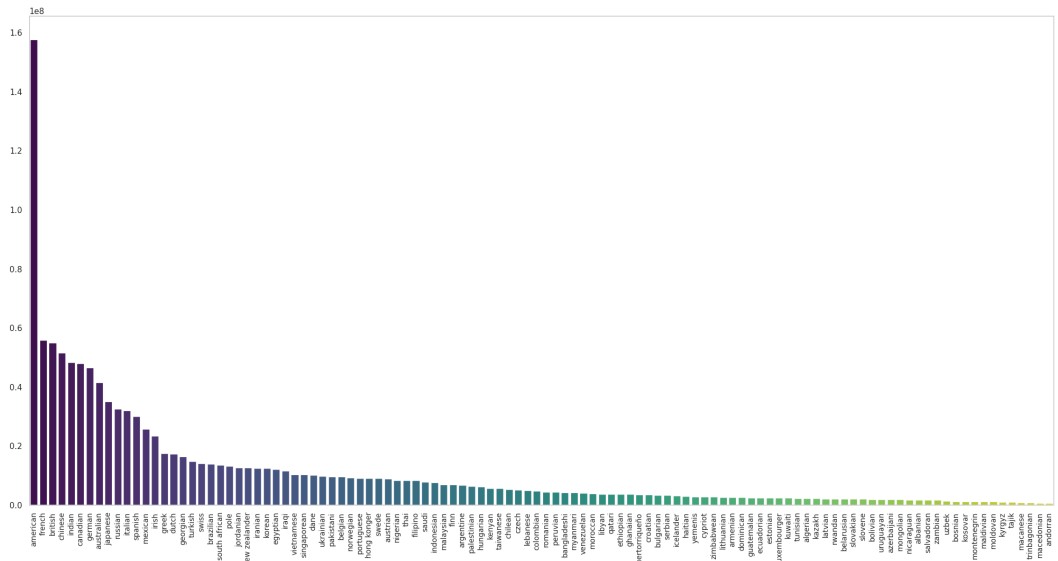

Figure 6: Number of documents in which the country is mentioned in RedPajama. The top 5 mentioned countries/regions are "American", "French", "British", "Chinese" and "Indian". The last 5 mentioned countries/regions are "Andorran", "Macedonian", "Trinbagonian", "Macanese" and "Tajik".

| Hit rate | | | | |
|---|---|---|---|---|
| | 17 | 70 | male | female |
| exercise | 48.3% (0.013) | 41.3% (0.011) | 55.9% (0.005) | 52.4% (0.018) |
| food | 51.9% (0.033) | 56.9% (0.035) | 62.0% (0.018) | 63.6% (0.028) |
| music | 42.4% (0.012) | 37.9% (0.016) | 45.0% (0.017) | 42.2% (0.009) |
| New rate | | | | |
| | 17 | 70 | male | female |
| exercise | 3.4% (0.0003) | 5.7% (0.001) | 6.8% (0.001) | 6.4% (0.002) |
| food | 5.5% (0.0004) | 5.1% (0.0008) | 6.3% (0.001) | 6.0% (0.001) |
| music | 22.8% (0.004) | 16.3% (0.004) | 21.0% (0.005) | 18.5% (0.005) |

Table 7: mistral-7b

| Hit rate | | | | |
|---|---|---|---|---|
| | 17 | 70 | male | female |
| exercise | 51.9% (0.038) | 44.8% (0.029) | 60.5% (0.034) | 68.2% (0.168) |
| food | 53.6% (0.015) | 55.2% (0.023) | 65.4% (0.015) | 69.6% (0.021) |
| music | 67.5% (0.014) | 64.3% (0.016) | 66.4% (0.003) | 56.5% (0.010) |
| New rate | | | | |
| | 17 | 70 | male | female |
| exercise | 4.3% (0.0005) | 6.1% (0.0009) | 9.5% (0.002) | 6.4% (0.001) |
| food | 3.4% (0.0002) | 2.7% (0.0001) | 5.5% (0.0003) | 5.0% (0.0002) |
| music | 18.6% (0.005) | 8.2% (0.001) | 13.7% (0.003) | 11.7% (0.004) |

Table 8: llama2-13b

## B.1 Ablation Studies on Demographics

For each model (llama2-13b and mistral-7b) and three representative topics (exercise routine, food, favorite music), we selected the culture with the most culture symbols extracted from each geographic region, totalling 12 cultures.

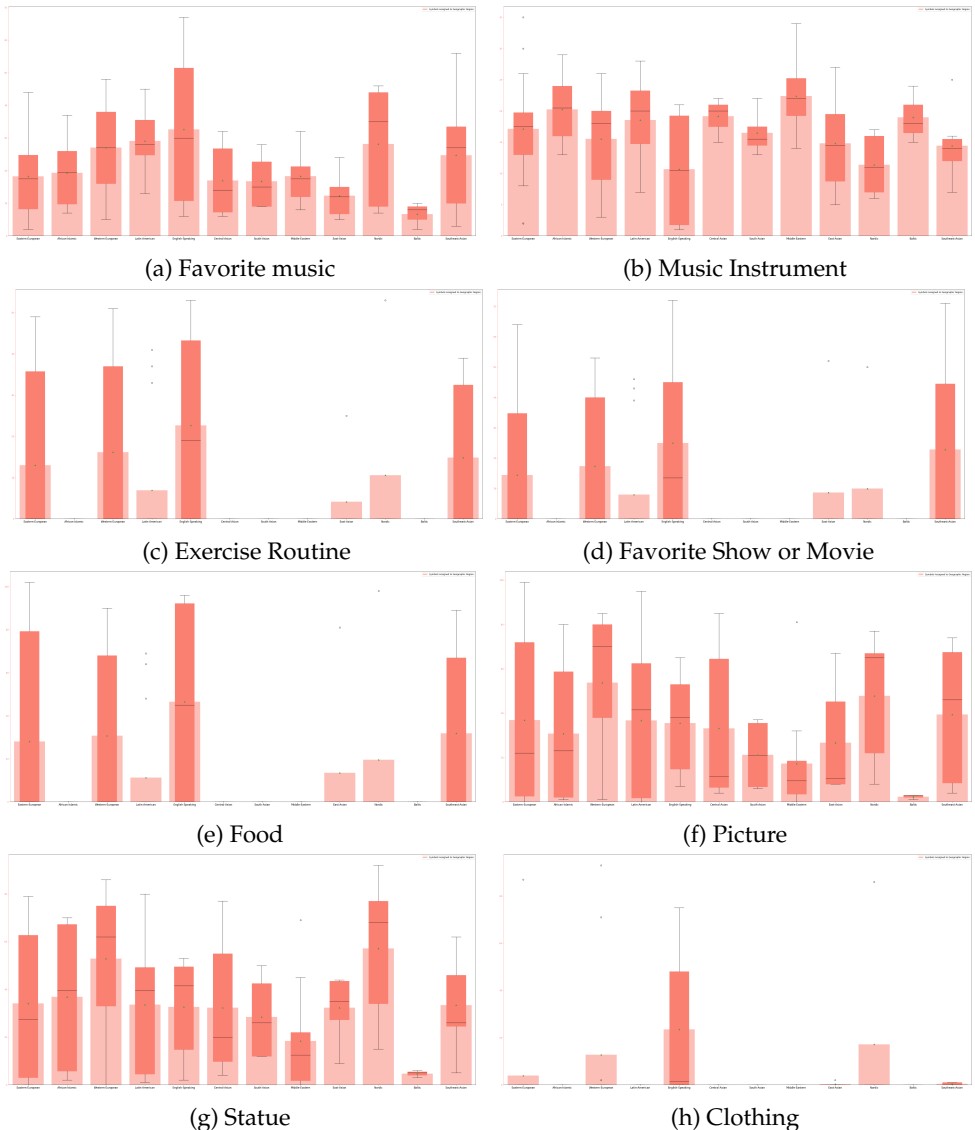

(a) Favorite music

(b) Music Instrument

(c) Exercise Routine

(d) Favorite Show or Movie

(e) Food

(f) Picture

(g) Statue

(h) Clothing

Figure 7: Geographic Region culture symbol extraction statistics for `llama2-13b`. From left to right, the geographic regions read: "Eastern-European", "African-Islamic", "Western-European", "Latin-American", "English-Speaking", "Central-Asian", "South-Asian", "Middle-Eastern", "East-Asian", "Nordic", "Baltic","Southeast-Asian".

For studying age variance, we tried 17 and 70 years old, representing younger and older generations, and prompted with *"Describe the [topic] of your neighbor. My neighbor is [culture] and is 17,70 years old. My neighbor probably …"*. For studying gender variances, we prompted with *"Describe the [topic] of your neighbor. My neighbor is [culture]. He, she probably …"*

In Table 7 and 8, we report the mean and average of hit rate, percentage of culture symbols extracted from neutral prompts that are generated in each condition, and the mean and average of new rate, percentage of generated symbols that are not a culture symbol previously detected.

We find that in both gender and age conditions for both models, models only generate a part of the culture symbols extracted from the neutral generations (as reported by hit rate). This suggests that by increasing the number of conditions, the generated culture symbols become more homogenous. The rest of the generations are either culture symbols that belong to

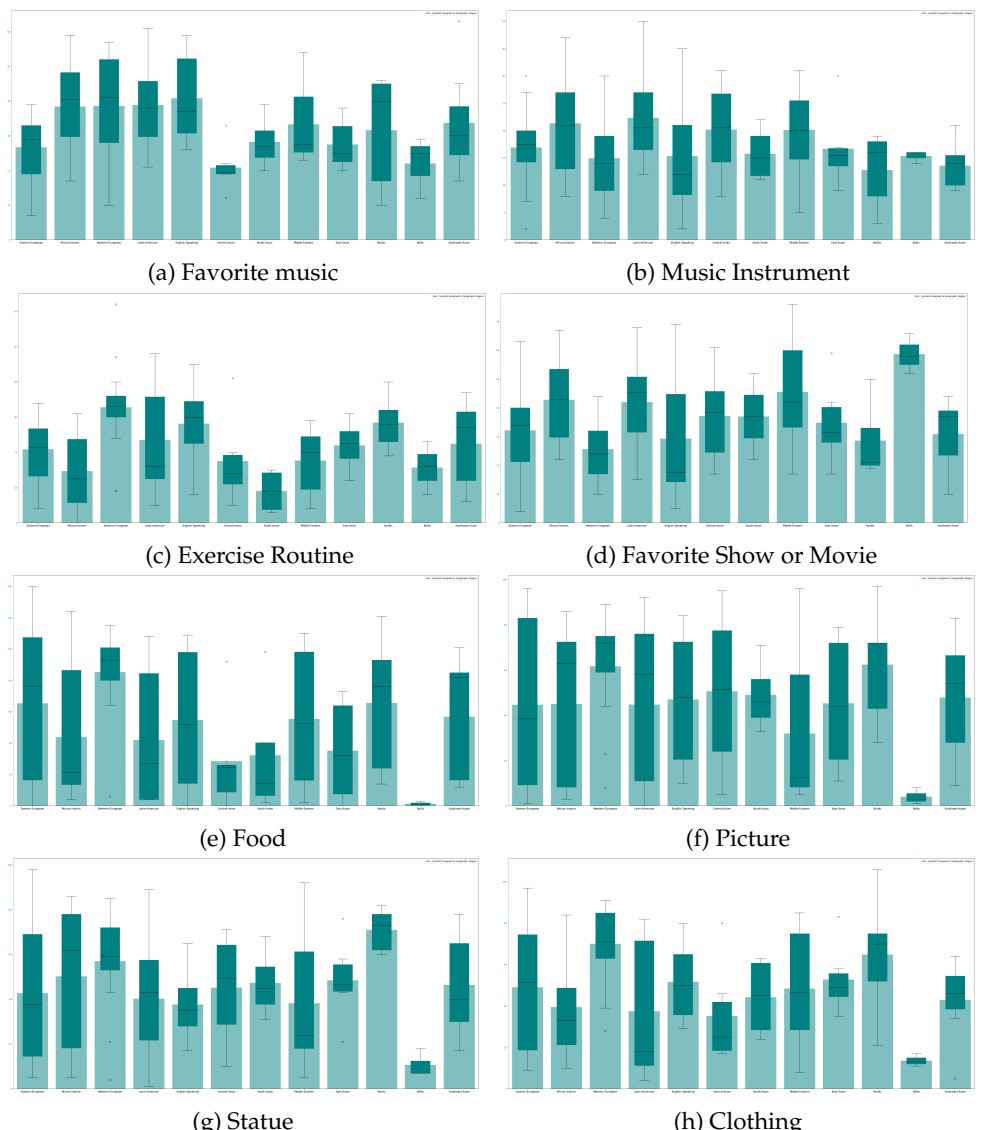

Figure 8: Geographic Region culture symbol extraction statistics for `mistral-7b`. From left to right, the geographic regions read: "Eastern-European", "African-Islamic", "Western-European", "Latin-American", "English-Speaking", "Central-Asian", "South-Asian", "Middle-Eastern", "East-Asian", "Nordic", "Baltic","Southeast-Asian".

other cultures, or symbols not previously recognized as culture symbols (as reported by new rate). The new rate shows that for exercise routines and food, there's only a small proportion of generations that have symbols unseen during neutral generations, and a slightly larger portion for music (understandable, as art genres differ more across subpopulations). We also would like to note that some of the newly discovered are not culture symbols, such as "another form of gentle exercise" in exercise routines of 70 year olds, but rather unique expressions, making the potentially unidentified culture symbols even fewer.

## B.2 Calibration of `mistral-7b` during unsupervised probability ranking.

Empirically, we find `mistral-7b` bias towards ranking a fixed set of cultures high on probability, which may result from the prior distribution of cultures in its training set. Therefore, we calibrate its probability with the probability of a sentence that does not contain any

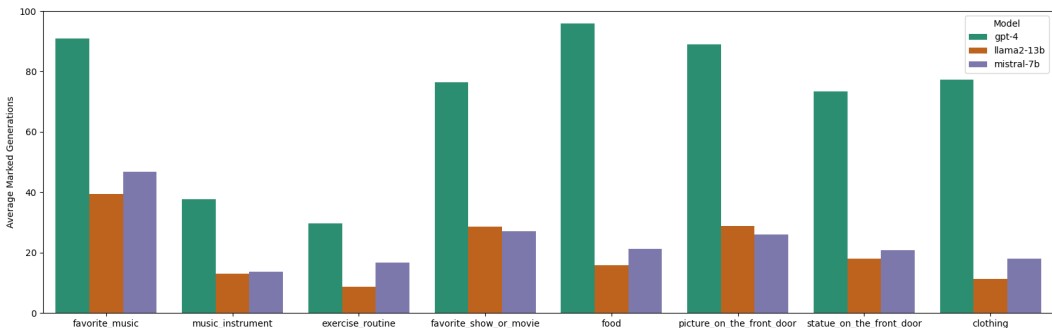

Figure 9: Average marked generations (out of 100 generations) by each model and topic.

culture symbols or nationalities, but only reflects the topic of interest, and we take a softmax over the calibrated the distribution. The calibration sentence for each topic is shown in Table 9.

| Topic | Prompt Template |
|---|---|
| favorite_music | My neighbor likes listening to music |
| music_instrument | My neighbor likes playing music instrument |
| exercise_routine | practices exercise |
| favorite_show_or_movie | My neighbor likes watching show or movie |
| food | For dinner, my neighbor probably likes to eat all kinds of food |
| picture | On the front door of the house, my neighbor has a picture |
| statue | On the front door of the house, my neighbor has a statue |
| clothing | My neighbor is wearing clothing |

Table 9: After each prompt, we append "My neighbor is [nationality]". We use the sentence probability to calibrate the sentence probability of `mistral-7b` on sentences with culture symbols, to even out the prior of culture distribution in a specific topic.

## C  Markedness

Figure 9 shows the average number of generations that contain either type of marker for each topic. `favorite_music` has the highest markedness of all topics, while `music_instrument` and `exercise_routine` have the lowest markedness. `gpt-4` has way higher markedness than `llama2-13b` or `mistral-7b`.

Table 10 shows the number of vocabulary and parentheses markers in culture-agnostic generations for each topic and each model. Compared to markedness of culture-conditioned generations, the default degree of markedness on culture-related topics is very low.

| | | favorite music | music instrument | exercise routine | favorite show or movie | food | picture | statue | clothing |
|---|---|---|---|---|---|---|---|---|---|
| llama2-13b | vocab | 0 | 2 | 1 | 0 | 0 | 0 | 0 | 0 |
| | paren | 0 | 2 | 0 | 1 | 2 | 2 | 1 | 1 |
| mistral-7b | vocab | 0 | 0 | 1 | 0 | 2 | 0 | 0 | 0 |
| | paren | 2 | 0 | 1 | 3 | 3 | 2 | 2 | 3 |
| gpt-4 | vocab | 0 | 0 | 0 | 0 | 10 | 1 | 1 | 0 |
| | paren | 0 | 0 | 0 | 0 | 0 | 0 | 0 | 0 |

Table 10: Markedness in culture-agnostic generations for all models.

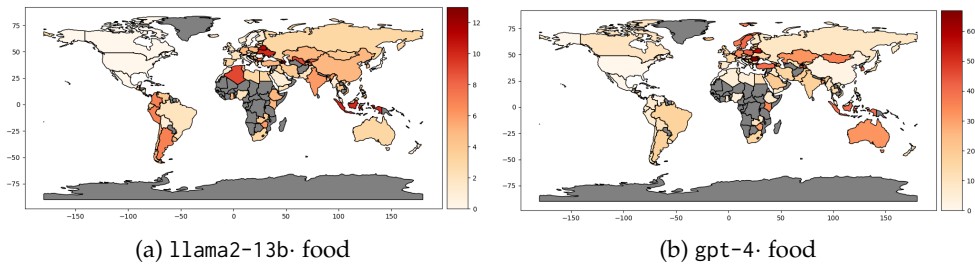

(a) `llama2-13b`· food              (b) `gpt-4`· food

Figure 10: Central Asia, East Europe and Southeast Asia have highest parentheses markers.

## D   Diversity

Geographic region shows significant difference, especially for `llama2-13b` on "Central-Asian", "South-Asian", "Middle-Eastern" cultures (Figure 7), and `mistral-7b` on "Baltic" cultures (Figure 8).

| Topic | Keywords |
|---|---|
| favorite_music | music, song, songs, album, albums, band, bands, singer, singers, musician, musicians, genre, genres, concert, concerts |
| music_instrument | music instrument, music instruments, instrument, instruments |
| exercise_routine | exercise, routine, workout, sport, sports |
| favorite_show_or_movie | movie, movies, film, films, TV show, TV shows, TV series, cinema |
| food | food, foods, cuisine, cuisines, dish, dishes, meal, meals, recipe, recipes, menu, menus, breakfast, lunch, dinner, snack, snacks |
| picture | picture, pictures, painting, paintings, portrait, portraits |
| statue | statue, statues, sculpture, sculptures |
| clothing | clothing, clothes, apparel, garment, garments, outfit, outfits, attire, attires, dress, dresses, suit, suits, uniform, uniforms |

Table 11: Keyword list that we use to measure topic-nationality co-occurrence frequency.

