# OpenReview forum: "CULTURE-GEN: Revealing Global Cultural Perception in Language Models through Natural Language Prompting"
_colmweb.org/COLM/2024/Conference — COLM_

### Official Review · Reviewer_bRDL · 2024-04-25

**Rating:** 7
**Confidence:** 5
**Ethics Flag:** 1

**Summary:**

In the recent past, there have been several papers that have explored the alignment of LLMs with human cultures, this paper takes a slightly different perspective. The paper explores how LLMs perceive global cultures and how does it reflect in their prompt specific generations.

In particular, authors prompt various LLMs on 8 different culture-related topics and from the generated responses culture symbols (culture specific topics) are extracted. Subsequently, the extracted culture symbols are classified (using unsupervised techniques) into corresponding region specific culture. This results in a new dataset CULTURE-GEN, having generated responses and culture symbols.

Using the above techniques and dataset, authors perform extensive set of experiments to evaluate LLMs and show the bias in LLMs towards dominant and popular cultures.

**Questions To Authors:**

Suggestions:
1. There are few typos that authors should fix, for example, second line: showed —> showed; second-last line of second paragraph; Page-4, the joint probability formula has ’n’ instead of ‘c’; Page 5 first line “We” —> “we”
2. In the related work, authors should also discuss more about proxies used for measure culture since it is hard to define culture. For example, authors can check out a recent survey:   https://arxiv.org/abs/2403.15412

**Reasons To Accept:**

1. Authors introduce a new dataset for examining LLM’s perception of global culture, this will be useful for the community.
2. Authors perform an extensive set of experiments to show bias in LLMs towards marginalized and non-mainstream cultures. To show this authors define “Cultural Markedness,” that covers both vocabulary-based and non-vocabulary-based markers that helps to distinguish between non-dominant cultures.

**Reasons To Reject:**

1. It is not clear how are cultural markers (vocabulary-based and non-vocabulary-based) obtained? I presume these were obtained via manual inspection of responses but this may not result in an exhaustive list of makers; how could one obtain more markers in an automated fashion, e.g., using LLMs?
2. While the paper performs in-depth analysis and experiments and show the bias in LLM’s perspective; the paper is lacking discussion on possible techniques to mitigate the bias; will standard alignment techniques based on RLHF/DPO suffice but in this case how does one resolve conflict between different cultures? Or one needs to develop a different model for each culture?
3. Authors are measuring culture via various topics but there has been recent work ( https://arxiv.org/abs/2403.15412 ) that argues that since culture is highly subjective, it is best to measure it via various proxies. Authors should discuss how do the topics align with various cultural proxies.

This is not necessarily weakness, but the paper has many typos that can easily be addressed.

---

> ### Author Rebuttal · Authors · 2024-05-30
>
> >W1: Automatic cultural markers discovery
>
> We did obtain the cultural markers from inspection. To automatically obtain markers, one may look for unigrams statistically different among cultures, such as (https://aclanthology.org/2023.acl-long.84.pdf). Their method requires a “marked”/“default” group which we do not have. To approximate, we used nationality pairs that are English-speaking/non English-speaking, eg. Canadian/Armenian, American/Albanian, on “favorite music”. We find that “traditional” is frequently in the top 10 words with the highest z-score, validating our inspection.
>
> Given our generations are < 10 tokens, it is unlikely to discover more cultural markers from them. Parentheses, however, can’t be discovered using word-based methods. For future work we can explore automated ways to discover cultural markers in long texts.
>
> >W2: Bias mitigation with RLHF
>
> We will add a discussion section about culture-bias mitigation. Our thoughts:
>
> Works (https://arxiv.org/pdf/2402.15018) show that monotonic alignment create disparities between default and marginalized groups and that reward models do not impact OOD behaviors as much as data. Hence, we need: 1. Expand global cultural coverage of data 2. Pluralistic alignment (https://arxiv.org/pdf/2402.05070). To resolve conflicts between different cultures, we decide the method based on the intended usage of the model:
>
> For general-purpose models, one may prefer models that output the whole spectrum of reasonable responses; for models for an audience in mind, one may prefer models that can be steered for attributions preferred by that group; for data generation models, one may prefer models who can generate an array of answers during sampling.
>
> >W3: cultural proxies
>
> Thank you for informing us of this contemporary work! We will add a paragraph in the discussion section. Our thoughts:
>
> The semantic proxies defined in this mentioned work mainly encompass words that have cross-lingual reference, while our work discovers entities that are unique to certain cultures. We try our best to match the topics to the defined proxies:
>
> picture, statue -> the house; food -> food and beverage; clothing -> clothing and grooming. A new category “practices and objects” may best fit “exercise routine”, “music”, “movie”, and “music instrument”.
>
> We used countries and regions as demographic proxies and we acknowledged in P4 1st paragraph that within one country there may be more granularly defined cultures.

---

> > ### Comment · Reviewer_bRDL · 2024-06-04
> > **Reponse**
> >
> > Thank you for your response.

---

### Official Review · Reviewer_icY8 · 2024-05-10

**Rating:** 7
**Confidence:** 3
**Ethics Flag:** 1

**Summary:**

Culture-Gen dataset: This work collects LLM outputs (generations) for 8 culture related topics for 110 countries using 3 different LLMs. Further, “culture symbols” are extracted from these outputs and matched to the associated cultures using an unsupervised sentence-probability ranking method.

Global Culture Perception/Cultural Fairness Evaluation: Two-fold evaluations first looks for “cultural markedness” (distinguishing non-default culture using vocabulary markers (e.g. the word traditional) and non-vocabulary markers (e.g. parenthesized explanations). The work further measures the diversity of the cultural knowledge by counting the number of symbols for a language and finding correlations with the RedPajama dataset count (to illustrate existing problems in training datasets). Finally they also look at preferred cultures in culture-agnostic generations.

**Questions To Authors:**

In section 4,
- I am not sure how you filter out the invalid phrases that do not contain any entities, like how do you decide that “traditional Albanian music” is invalid whereas “songs by Vitas” is valid.
- In the next paragraph, you mean P(c, e|T) right? There's likely a typo.

**Reasons To Accept:**

- Diverse data set and automated method of collection of data that is potentially going to be useful for data collection at geographical levels smaller than a country.
- Important study that looks at markedness in generations. This is something that I have noticed as well in my experience with these models, so it is good to see this phenomenon being studied and quantified.
- An interesting exploration of the correlation to document counts in RedPajama to quantify claims made about training data instead of just claiming that the effects can be explained by training data. Motivating usage of OLMo and other such models with open sourced data is hopefully going to inspire others working on this to explore these problems and come up with solutions.

**Reasons To Reject:**

There is a trade-off between defining cultural symbols accurately versus scraping it from LLMs, as discussed in Section 6. But I wonder whether generating data from sources that you find to be “biased” might have inherent issues in the design?

---

> ### Author Rebuttal · Authors · 2024-05-30
>
> > W1: Collecting data from biased LLMs
>
> Generating data from LLMs for any purpose not limited to bias studies all face similar concerns in correctness, diversity, and hallucination[1]. For other works that aim to collect comprehensive cultural data from LLMs and use it in downstream training or tuning language models, one will need to take extra care to mitigate bias in the data. However, as explained in the paragraph after the one we discussed the trade-off, our work focuses on analyzing the cultural perceptions of the LLMs, and does not intend the cultural symbols to be used in downstream training. That’s why we chose the continue-generation task and expected the result to be biased.
>
> [1] Guo, Xu, and Yiqiang Chen. "Generative AI for Synthetic Data Generation: Methods, Challenges and the Future."
>
> > Q1: filter out the invalid phrases
>
> We deem “traditional Albanian music” as invalid, because this is a boilerplate answer that can be applied to all cultures, such as “traditional Algerian music”, “traditional Nigerian music”, etc. So we will filter for culture symbol candidates that contain “traditional” or “[culture]”, where [culture] is the culture for which this phrase is generated. Therefore, if for culture A the model generates “[culture B] music,” we will keep the answers.
>
> >Q2: Typo
>
> Thanks for catching that! Will fix.

---

> > ### Comment · Reviewer_icY8 · 2024-06-07
> > **Acknowledging the rebuttal**
> >
> > Thank you for your responses!

---

### Official Review · Reviewer_5nKe · 2024-05-10

**Rating:** 7
**Confidence:** 3
**Ethics Flag:** 1

**Summary:**

In this paper, the authors have developed natural language prompts to elicit language model perceptions of different cultures (through cultural symbols such as clothing, food, etc.). This study discovers that culture-conditioned generation (i.e., priming the prompt with the phrase - My neighbor is <nationality>) generates more text with othering markers (such as traditional) for less-represented cultures compared to default cultures.

**Questions To Authors:**

1. Table 1 - The prompts designed for each topic are ambiguous. For instance, playing could also be in context of sports, theater etc. Likewise for practices (practice archery, practice guitar etc.). I tried these prompts on GPT-4 and the generated text do include these domains. How do the authors ensure that extracted cultural symbols belong to specified topics?

 2. The authors have assumed that all cultures have a picture or statue on the front door - please provide source behind this assumption.

 3. I am also curios to know how representative these cultural symbols are - for instance, favorite show or movie - how is the distribution across age and gender? I tried prompts in Table 5 - I asked model to predict age and gender along with culture. Mostly, it suggests "younger demographic".

 4. Does the word "traditional" always indicate markedness (in sense of otherness)? There is a wikipage on traditional folk music- https://en.wikipedia.org/wiki/Folk_music#Traditional_folk_music
In Appendix C, the authors also noted this markedness is more common in music category. The same could be said for clothing ("traditional dress or clothing").

 5. Pg 6 - "Table 7 and 8 show the distribution of culture symbols for each geographic region" - Table should be Fig. Some geographic regions have many more countries (e.g., Eastern Europe vs Baltic). How does this influence the outcome (#culture symbols, markedness)? Was any kind of normalisation performed?

**Reasons To Accept:**

The study helps understand language model perceptions of national cultures - a step ahead of Western-Eastern cultural studies.

**Reasons To Reject:**

The major contribution of this study is analyzing language models' perception toward cultures and possible, othering of marginalized cultures. Both of these aspects need more clarity. Please see questions.

---

> ### Author Rebuttal · Authors · 2024-05-30
>
> > Q1: Table 1
>
> See P4 2nd paragraph.
>
> > Q2: representativeness of cultural symbols
>
> We were not studying what a language model *can* generate, but what a language model *will* generate. It is expected that the current set of generations show bias towards certain subpopulations. We will study subpopulation biases in future work.
>
> We tried mistral-7b on 3 topics with the culture with most symbols in each geographic region.
>
> We report hit rate, avg percentage of collected culture symbols that are generated in each condition, and new rate, avg percentage of generated symbols that are not culture symbols previously detected.
>
> | H        | 17yo | 70yo | male | female |
> |----------|------|------|------|--------|
> | exercise | 48.3 | 41.3 | 55.9 |  52.4  |
> | food     | 51.9 | 56.9 | 62.0 |  63.6  |
> | music    | 42.4 | 37.9 | 45.0 |  42.2  |
>
> | N        | 17yo | 70yo | male | female |
> |----------|------|------|------|--------|
> | exercise |  3.4 |  5.7 |  6.8 |    6.4 |
> | food     |  5.5 |  5.1 |  6.3 |    6.0 |
> | music    | 22.8 | 16.3 | 21.0 |   18.5 |
>
> TLDR: quite representative:
>
> Model only generates part of all culture symbols extracted (hit rate), i.e. the generated culture symbols become more homogenous.
> Only a tiny part of generations have unseen symbols (new rate). Note that some of these symbols are not culture symbols, rather unique expressions.
>
> > Q3: picture & statue
>
> For pictures and statues, we intended to extract culture symbols that do not belong to the other topics but still have cultural significance, eg. animals, religious symbols and historical figures. In downstream tasks like story generation, one may ask the model to describe decorations containing objects of cultural significance, but those objects may be hard to categorize.
>
> > Q4: traditional & markedness
>
> “Traditional music” or “traditional clothes” may overall increase the frequency of “traditional”, but the difference in model usage of these combinations manifests markedness. For the “default” cultures, “traditional” is not the top choice in a model’s decoding process of the first word. As low as 1% GPT-4 generations contain markers for those cultures.
>
> > Q5: geographic regions
>
> We report averaged metrics for each region as normalization.
>
> The number of countries in each geographic region will impact the variance of the # culture symbols. The box plots show high IQR regions, for which highlighted the importance of studying variance within each region.

---

> > ### Comment · Reviewer_5nKe · 2024-05-31
> >
> > Thank you for your detailed response. I have updated the score based on authors' rebuttal.

---

> > > ### Author Response · Authors · 2024-05-31
> > >
> > > Thank you so much for recognizing the contribution of our work! We really appreciated your detailed and informative feedback.

---

### Official Review · Reviewer_rAD4 · 2024-05-11

**Rating:** 7
**Confidence:** 3
**Ethics Flag:** 1

**Summary:**

this works proposes a framework to assess the fair perception of cultural aspects on LLMs, with criterias ranging over-representation,  diversity and the idea of a `global culture` .  This is operationalized by a generation phase (based on custom prompts to obtain sentences dealing with a set of specific topics conditioned by country)
followed by a `culture topic` extraction phase.
Then there are requirements specific to to the concept of global culture perception in terms of diversity and fairness
The combination both the dataset and the analytical framework allows to undercover a given LLM cultural perception.

Results are in some sense what we may expect, given what we know (or dont know) about the training datasets used for the target LLMs under study : there are indeed cultures that are mariginalized and also knowledge is in some way directly proportional to frequency metrics

**Questions To Authors:**

- is there any way to incorporate the audience into the prompting? For example, i wonder of the emergence of markedness, specifically the use of parenthesis changes if i specify more information . In other words, what conditions the LLMs to provide detailed explanation about a certain topic/ term ? What if the prompt is like "I'm (European/Asian) . My neighbor is ..."  Would the result change based on the extra information im providing (LLM may assume i *know* already about a topic)

- i could be missing some part but, is it possible to obtain an estimation for the markedness in the default case of English ? I guess the purpose of parenthesis usage would be more towards clarifying a topic that is complex and is assumed most people would not know.

- while probably not in the scope of the current submission, but is there anything to add about using LLMs that are not only English-based ? How accentuated could be the results ? Im wondering there could be differences as language usage would be more grounded

**Reasons To Accept:**

The topic is inherently relevant to the community, and has big potential in the sense of analyzing LLMs from a more holistic, broad perspective.

The framework and its components and steps are simple and well justified, specially in terms of the decisions associated to target countries and also de definitions of culture topics (and the prompt design strategies to obtain them).

There is an explicit section that discusses the impact of the training data . I consider this makes a different with other submissions under the same topic. I appreciate that while we don't have access to the actual training data , the authors at least made the effort to acknowledge and even approximate its impact .

**Reasons To Reject:**

The analysis is based on enumeration of possible cases/ scenarios. Therefore it could be hard to assess it generalization power (would need to exhaustively go through all possible cases, attribute combinations )

---

> ### Author Rebuttal · Authors · 2024-05-30
>
> >Q1: Incorporate the audience into the prompting.
>
> We quickly tried this prompt: “Describe the favorite music of your neighbor. I am Azerbaijani. My neighbor is Azerbaijani. My neighbor probably likes listening to” and asked GPT-4 to generate 10 times. Only one of them has parentheses: “Azerbaijani traditional folk music, Meykhana (a traditional Azerbaijani music characterized by catchy rhythm and spontaneous lyrics), Mugham (a”. Seems like adding an audience from the same nationality does not completely remove markedness, although it may reduce the frequency that it occurs.
>
> We will add a discussion paragraph and a more systematic experiment in the camera-ready version.
>
> >Q2: estimation for the markedness.
>
> We examined the markedness of culture-agnostic generations, and found that GPT-4 has 0 vocabulary or parentheses markedness for most topics (exceptions are picture and statue, with only 1 generation each), llama2-13b has as many as 2 vocabulary and parentheses markedness among all topics, and mistral-7b has as many as 3 parentheses markedness for 3 out of 8 topics: markedness behavior does not appear when the model is only asked to talk about a certain culture.
>
> One could count the frequency of parentheses in an indexed pre-training data, similar to what we did in 5.2. But our indexing server does not index punctuations, so we cannot get an estimate for parentheses. For reference, # documents with the word “traditional” is 51563138, 0.25% out of 20B documents of RedPajama-v2, quite a big portion.
>
> We will add these two results to the camera-ready version.
>
> >Q3: LLMs that are not only English-based.
>
> This is indeed out of scope of the current work, but it is an interesting topic that previous works have touched upon. [1] evaluated multilingual models and monolingual Arabic models for cultural biases, and found that even monolingual Arabic-specific LMs exhibit Western bias, potentially because part of the pretraining-data, albeit solely in Arabic, often discusses Western topics. They also found that multilingual LMs show even stronger Western bias, suggesting that multilingual training could impact cultural relevance of non-western languages.
>
> Based on the takeaways from previous works, we speculate that there are shared trends between English-only and non-English models, but we will defer the accurate study on this topic for another work.
>
> [1] Naous, Tarek, et al. "Having beer after prayer? measuring cultural bias in large language models."

---

> > ### Comment · Reviewer_rAD4 · 2024-06-05
> >
> > Thank you for your detailed answer and for putting the time to perform extra experiments.
> >
> > -Regarding Q1. I think that is a feasible way to proceed. Nevertheless, i could not be easy to guarantee the generality of the reuslts as most likely you will end up in a combinatorial problem. Another interesting example could be to use a negation (I am NOT Azerbaijani ... )

---

> > > ### Author Response · Authors · 2024-06-06
> > >
> > > Thank you for your suggestion! Using negation makes a lot of sense. We will add that to our camera ready experiments too.

---

### Decision · Program_Chairs · 2024-07-10

**Decision:**

Accept

**Comment:**

The paper presents a novel framework to evaluate the cultural aspects of LLMs via prompting. The study focuses on assessing over-representation, diversity, and the concept of a global culture by analyzing generated text based on country-specific prompts.

This work received unanimously positive reviews. The reviewers agree that this work is highly relevant to the community, and the proposed framework is well-justified. This work also produces a new dataset, which reviewers agree is useful. There are some clarity issues in the writing that can be fixed in the camera-ready version. Overall, I recommend accepting this paper.